# Driving Behavior That Limits Concentration: A Nationwide Survey in Greece

**DOI:** 10.3390/ijerph18084104

**Published:** 2021-04-13

**Authors:** Anna Tzortzi, Melpo Kapetanstrataki, Vaso Evangelopoulou, Panagiotis Behrakis

**Affiliations:** 1George D. Behrakis Research Lab, Hellenic Cancer Society, 10557 Athens, Greece; atzortzi@researchlab.gr (A.T.); vevan@researchlab.gr (V.E.); pbehrakis@acg.edu (P.B.); 2Institute of Public Health, The American College of Greece, 10557 Athens, Greece; 3Athens Medical Center, Distomou 5-7, Marousi, 15125 Athens, Greece

**Keywords:** driving behavior, limited concentration driving, drink driving, smoking and driving, texting, GPS setting, alcohol and driving, distractive behavior, road accidents, traffic safety

## Abstract

Human behavior is implicated in most road accidents. The current study examined drivers’ behavior that interferes with decision making and reaction time to an incidence. Adults (≥17 years-old) participated in a questionnaire-based survey for driver’s behavior. Dataset was weighed according to sex, age and education based on the 2011 census. Differences between groups were assessed with Chi-squared tests while logistic regression models were used to identify drivers’ characteristics for specific behaviors. A total 1601 adults participated in the survey—48% males and 52% females. Texting, Global Positioning System (GPS) setting and smoking were observed more by professional drivers and drivers of an urban area, while smoking was also dependent on social class. Drink driving was observed more by males (20% vs. 5% females), while after adjusting for age, the odds of drink driving in males were 5 times higher than females (*p* < 0.001). A different effect of age depending on the driver’s sex and vice versa was observed regarding phone calls. Drivers’ behavior with distractive potential differed by age, sex, social class and area of residence. Male drivers were more likely to perform drink driving, while professional drivers were more likely to use cell phone for calls and texting, set the GPS and smoke while driving.

## 1. Introduction

Driving is a demanding act that requires constant attention and fast processing of an abundance of information for instant decision making and action. Mental, emotional and physical state greatly interfere with the ability, speed and quality of the decision process itself, where the intellectual functions, judgment, preference and choice are integrated into shaping the final decision and instantaneously transform it into action [1].

Ethnicity, civilization, age and gender, factors known to shape behavior [2], potentially predict some degree of diversity in driving behavior as well.

Traffic and traffic safety is the result of the continuous interplay between four major factors arising from the road, vehicle, environmental conditions and the human user. Human behavior, the most unpredictable of the three factors, has been extensively studied, especially so in recent years in the frame of smart cars and automotive control technology development [3]. 

In the majority of road accidents, factors related to human behavior have been identified as causally implicated, and therefore they are totally preventable [4]. Although certain behavioral factors have been regulated by specific laws and the driving code (i.e., alcohol, cell phone), others have been less studied as potential road accident factors, as is the case with sleep disorders [5], opioid, antipsychotic and antidepressant medication, in contrast to their extensively studied medical aspect [6]. 

Greece has succeeded a significant reduction in road accidents, approaching the European Union (EU) average for passenger car accidents (24.9 vs. 23.5 fatalities/million inhabitants), as per 2018 data. It is, however, double the mortality due to motorcyclist accidents compared to the EU average (17.7 fatalities/million inhabitants in Greece vs. 7.9 fatalities/million inhabitants in the EU) [7]. These data show there is still room for improvement and an urgent need to tackle the respective burdens more efficiently.

Studies that have examined drivers’ behavior in Greece are scarce and limited. Behaviors include specific factors such as mobile phone use [8], driving with mild cognitive impairment [9], disobedient driving of epileptic patients [10], as well as the personality and driving attitudes of bus drivers [11]. The most inclusive survey that has examined the reported behavior, attitude and opinion on traffic behavior, traffic law enforcement and policy measures of all road users (drivers, cyclists, pedestrians), is the ESRA2 (E-Survey of Road Users’ Attitudes), conducted in 2018 with the participation of Greece and other countries [12]. The most recently published study by Yannis et al., based on the same ESRA2 data, focused on the attitudes of vulnerable road users (pedestrians, cyclists and motorcyclists) [13].

To the author’s knowledge, there is a gap in recent studies specifically focusing on drivers’ distractive behavior in Greece. Thus, we expand beyond the obvious restrictive policy and law enforcement indicators, to consider additional traffic accident factors related to certain lifestyle choices, as well as health and physical state.

Therefore, the aim of the current study was to examine drivers’ reported behavior with potential to limit concentration and interfere with decision making, such as cell phone use, texting, Global Positioning System (GPS) setting, smoking while driving, drink driving, driving under the influence of drowsiness caused by medication, irritation and sleep deprivation. Furthermore, the study aimed to profile the driver who is more prone to exhibit the abovementioned behavior. 

## 2. Materials and Methods

### 2.1. Sample 

Adult Greek residents, across all the Greek territory, aged 17 years-old and above at the time of the survey were enrolled.

### 2.2. Survey Timeframe 

Data collection took place in September 2020.

### 2.3. Data Collection

Data collection was conducted by computer assisted telephone interviews (CATI) with the use of a structured questionnaire. Random digital dialing was performed using residence areas quotas defined by the Nomenclature of Territorial Units for Statistics II (NUTSII) classification. 

Interviews were performed by Metron Analysis, a research company in the fields of social, political and market research areas. 

Informed consent was obtained verbally by each respondent prior to the onset of the questionnaire, at the beginning of the call. Individuals were informed about the scope and duration of the survey, the organization conducting the survey and the way their data were to be handled: fully anonymized and only for statistical purposes. Then they were asked to give their verbal consent to participate in the survey; those who opted in continued by answering the questionnaire, whereas those who opted out were thanked for their time and the call was ended. Individuals who wanted to participate in a different time of the day were given the option to book another time slot.

The response rate was 15.2% and a 95% sampling error was calculated as ±2.5%, assuming random sampling methodology.

### 2.4. Validation

A total 21% of interviews were validated by co-listening the interview in process and interviews were ended if a participant seemed to respond at random or when the participant declared they were not interested or too tired to continue with the interview. Only fully completed questionnaires were included in the analysis. All questionnaires were checked before a fully anonymized dataset was passed onto our team for analysis and results dissemination.

### 2.5. Questionnaire

The questionnaire consisted of forty-three questions (both closed-ended and open-ended) that lasted approximately fourteen minutes per participant. It was structured in five sections as shown in Table 1. 

The current study examined questions of Sections #1, #2 and #5. Participants responded to questions seeking behavioral characteristics based on their memory recollection for the past three months’ driving.

### 2.6. Dataset Weighing

The demographic distribution of our dataset was compared with that of the 2011 census, performed by the Hellenic Statistical Authority and the dataset was weighed according to sex, age and education. 

### 2.7. Statistical Analysis

Analysis was performed using the calculated weights as the frequency variable. Normality of continuous variables was assessed with the Shapiro–Wilk statistic. For categorical variables, comparisons between groups were performed with the Chi-squared test. The Spearman correlation coefficient was used to assess correlations between behavioural variables. Results are focused on differences between demographic characteristics and are presented as frequencies or percentages for categorical variables, whereas for continuous, non-normally distributed variables, the median and the interquartile range (IQR) are presented. 

Logistic regression models were performed looking into behavior that limits concentration from driving and drivers’ characteristics. The initial models included sex and age, the interaction of sex and age as well as all variables deemed statistically significant in the univariate analyses. Backward elimination procedure was used to conclude to the final models.

Statistical significance was set at *p* < 0.05. All *p*-values presented are two-tailed. Analysis was performed in Stata 14 (StataCorp. 2015. Stata Statistical Software: Release 14. College Station, TX, USA: StataCorp LP). Figures were created in Microsoft Excel (Office 365). 

### 2.8. Specification of Variables

The social class variable was calculated based on information provided regarding the highest educational level and occupation of the main provider of the household.

The term ‘drink driving’ in the manuscript refers to participants that drove after having drunk more than one drink. 

The term smoking in the manuscript refers to any tobacco product used.

## 3. Results

### 3.1. Main Findings 

A total 1601 adults, 17 years-old and above participated in the survey, with 48% males and 52% females. Of these, 74% had a driving license, whereas 26% did not. Demographic and driver’s characteristics are presented in detail in Table 2 and Table 3. 

Regarding behavior that limits concentration while driving, 95% of participants declared reacting at least once to other drivers’ behavior that caused them to be irritated, 49% declared talking on the phone, 20% declared smoking and 18% declared setting their GPS while they were en route. Finally, 13% admitted to drink driving at least once, while 10% were texting while driving (Table 4). 

Since only 1% admitted driving after taking medication causing drowsiness, results were not further analyzed.

Less than half of the participants (46%) declared always getting an 8-h sleep before a long journey while 20% only sometimes do (Figure 1).

### 3.2. Univariate Analyses 

#### 3.2.1. Type of Vehicle

Cell phone calls (*p* < 0.001), smoking (<0.001), setting the GPS (*p* = 0.002) and texting (*p* < 0.001) while driving were observed more in professional compared to private car and motorbike drivers (Table 4).

#### 3.2.2. Sex

Females not owning a driving license were more than double of males, while those owning two licenses were 6 times less compared to males (*p* < 0.001). The vast majority of females drove a car in the past three months while a small proportion drove a motorbike, compared to less male car drivers and more motorbike drivers (*p* < 0.001) (Table 5). 

A highly statistically significant difference was observed between sexes and drink driving—20% of males compared to 5% of females (*p* < 0.001) (Table 6).

Highly statistically significant differences were observed regarding 8-h sleep before a long journey between sexes, with more males than females getting an 8-h sleep before a long journey (*p* = 0.002) (Figure 1).

#### 3.2.3. Age

Differences in driver’s characteristics among age groups are shown in Table 5.

Reacting to other drivers’ irritating behavior was observed more in younger individuals compared to older aged individuals, differences highly statistically significant (*p* < 0.001). Cell phone calls (*p* < 0.001) and smoking (*p* = 0.0005) were observed more in 35–54-year-olds compared to the rest age groups while it was observed the least in 75+ year-olds. Texting (*p* < 0.0010 and setting the GPS (*p* < 0.001) while driving were observed more in younger individuals (Table 6).

Highly statistically significant differences were observed between different age groups and getting 8-h sleep before a long journey, with more individuals 17–34-years-old declaring that they always and often do, compared to other age groups (*p* < 0.001) (Figure 1).

#### 3.2.4. Educational Level

More individuals with a higher education had at least one driving license compared to individuals with an education up to secondary (*p* < 0.001) (Table 5).

Cell phone calls (*p* = 0.004), setting the GPS (*p* = 0.0009) and texting (*p* = 0.0002) while driving were observed in a higher proportion in individuals with a higher educational level compared to those with an up to secondary education (Table 6). 

More individuals of a higher education get an 8-h sleep before a long journey compared to individuals of an up to secondary (*p* < 0.001) (Figure 1). 

#### 3.2.5. Social Class

A higher proportion of participants of an upper and a middle to upper social class had at least one driving license compared to individuals of a middle to lower and a lower social class (*p* < 0.001). Furthermore, a lower proportion of individuals of an upper social class drove mostly a motorbike in the past three months, while a higher proportion of participants of a middle to lower and a lower social class drove mostly a professional vehicle (*p* = 0.01) (Table 5).

Reacting to other drivers’ irritating behavior (*p* = 0.03) and cell phone calls while driving (*p* = 0.001) was observed less in participants of a lower social class compared to participants of the other social classes. Smoking while driving was observed more in lower-class individuals compared to those of an upper class, while those of a middle-class had a similar proportion (*p* = 0.02). Finally, texting while driving was observed more by individuals of a middle to lower social class compared to the other social classes (*p* = 0.02) (Table 6).

#### 3.2.6. Area of Residence

No differences were observed between participants living in different kind of areas and the number of driving licenses they have or the vehicle they mostly drove in the past three months (Table 5).

Setting the GPS while driving (*p* = 0.03) was observed more in participants living in urban areas and towns than rural areas, while texting was observed more in participants living in urban areas compared to towns and rural areas (*p* = 0.01) (Table 6).

### 3.3. Correlations between Variables

Most correlations between behaviors that limit concentration were weak to non-existent. A moderate to weak correlation was observed between texting and setting the GPS while driving (rho = 0.36, *p* < 0.001) (Appendix A).

### 3.4. Multivariate Analyses 

#### 3.4.1. Drink Driving 

Logistic regression analysis showed that adjusting for age, males had five times higher odds of drink driving than females, a difference that was highly statistically significant (*p* < 0.001). Additionally, adjusting for sex, individuals aged 17–34-years-old had 8 times higher odds of drink driving than those aged 75+ (*p* = 0.002), 35–54-year-olds had 6.2 times higher odds than those aged 75+ (*p* = 0.005) and 55–74-year-olds had 6.8 times higher odds than those aged 75+ years-old (*p* = 0.003) (Table 7).

#### 3.4.2. Cell Phone Calls

Adjusted for age and sex, private car drivers had 5.6 times higher odds of talking on the phone while driving compared to motorbike drivers (*p* < 0.001), while for those driving a professional vehicle the odds were 6.8 times higher (*p* < 0.001). Furthermore, an interaction was observed between age and sex, showing that the effect of age on talking on the phone while driving depends on the drivers’ sex and vice versa. In particular, males aged 55–74-years-old, compared to females, had three times higher odds of talking on the phone while driving, adjusted for the type of vehicle driven (Table 7). 

#### 3.4.3. Smoke Driving

Males had 1.4 times higher odds of smoking while driving than females (*p* = 0.043). Drivers 35–54 years-old were 2 times more likely to smoke while driving than 17–34-year-olds (*p* = 0.009) while 17–34-year-olds were 6 times more likely to smoke compared to 75+ year-olds (*p* = 0.006). Private car drivers were 3.6 times more likely to smoke than motorbike drivers (*p* = 0.002) while professional drivers were 9.5 times more likely to smoke (*p* < 0.001). Drivers of a lower social class had 3.7 times higher odds of smoking compared to drivers of an upper social class (*p* < 0.001). Finally, compared to rural areas, residents of an urban area (*p* = 0.002) and a town (*p* = 0.013) were three times more likely to smoke. Results were adjusted for sex, age, driven vehicle, social class and area of residence (Table 7).

#### 3.4.4. Texting and/or Setting the GPS While Driving

Drivers 17–34-years-old were 9.1 times more likely to text/set their GPS while driving compared to 55–74-year-olds (*p* < 0.001), whereas 35–54-year-olds were 4 times more likely (*p* < 0.001). Compared to motorbike drivers, private car drivers were 5 times more likely (*p* = 0.003) and professional drivers were 9.2 times more likely (*p* < 0.001) to text/set their GPS while driving. Finally, residents of an urban area were 2.7 times more likely to text/set their GPS while driving compared to residents of a rural area (*p* = 0.008). Results were adjusted for sex, age, driven vehicle and area of residence (Table 7).

## 4. Discussion

To the authors knowledge, this is the most recent study to examine drivers’ reported behavioral characteristics that interfere with the ability to effectively focus on the act of driving in Greece. It was also the first study to include smoking/vaping among distractive driving behaviors.

Causes for distraction and/or impaired concentration identified by the present study, were cell phone use, text messaging, GPS setting, drink driving, smoking while driving, driving under the influence of drowsiness caused by medication and driving while being irritated or sleep deprived. Interestingly, the present study found most of the aforementioned behavior characteristics to greatly depend on the driver’s age and sex. 

### 4.1. Alcohol

In the current study, males and younger individuals were more likely to exhibit drink-driving than women and older drivers respectively. 

According to the recent health survey by the Hellenic Statistical Authority [14], 6% of those over 15 years old consume alcohol on a daily basis, among them adolescents and young adults. Alcohol use in adolescence is associated with cognitive impairment that may continue through adulthood or may lead to dependence [15]. In the present study, far less women reported drink driving than men. However, as shown by previous studies, the alcohol effect is greater in females, possibly due to their increased sensitivity, different glucose metabolism, in addition to hormonal, body size and constitution (more fat tissue and water) related reasons [16].

While the legal alcohol limit in Greece is set at 0.25 mg/lt exhaled breath (BrAC) and 0.50 mg/lt blood (blood alcohol concentration—BAC) [17], police reports in 2019, showed that 90% of the BrAC tests given to drivers engaged in an accident, were found within normal (legal) limits [18], a finding that warrants a revision of this “legally allowed drink driving.” Furthermore, police data show an increased rate of fatal accidents on Friday and Sunday [18], a fact likely explained by the alcohol ingestion on weekend night out and/or long travels.

According to a previous simulation study, alcohol impairs driver’s cognitive performance leading to a delayed reaction to an incident, increased impulsiveness and impaired braking/accelerating behavior, even at 0.03% BAC, concentrations corresponding to the legally set maximum driving limit in some countries. The same study also showed that alcohol affects female driving performance at even lower BAC concentrations compared with males who were affected in concentrations higher than 0.08% [19]. However, young male drivers have been shown more likely to commit aggressive driving (higher speed and acceleration) after having consumed alcohol [20]. Paleti et al., in their analysis of post-accident data for the influence of aggressive driving, found that naïve, adolescent drivers, without driving license, who drink drive, do not use a seat belt and drive a pick-up truck are more likely to commit aggressive driving. The study is based on USA data where adolescents are legally permitted to drive since their 16th year of age [21]. According to the present study 17–34-years-old have higher odds of drink driving, a finding in line with the aforementioned study by Paleti et al., except for the age range of 16–20-years-old that does not entirely fit the Greek affairs, as in Greece, eligibility for licensed driving requires individuals to be 18 years old.

The combination of alcohol, even at low concentrations, and mild distractions develops a more demanding situation, that enhances the individually induced driving performance impairment due to each factor and further increases the risk to engage in an accident [22]. 

Moreover, the combination of alcohol and other medication, or the combined effect of complex therapeutic regimens on cognitive status and driving behavior are not extensively studied, highlighting a gap to be addressed in the contemporary multi-medicated members of human society [6]. 

### 4.2. Cell Phone/ Texting/GPS Setting

Almost half the drivers in the current study reported using cell phone while driving, a finding in line with ESRA2 survey for Greece [12]. Professional and private drivers were found in the current study significantly more likely to report cell phone use while driving, than motorbike drivers. Studies have shown that road accidents are nine times more likely when cell phone is used while driving [23]. 

To examine the impact of cell phone on driver’s behavior, Zhang et al. divided phone use in distinct functions (dial, answer, talk, listen, hang-up and information view) and then analyzed the impact on driving control behavior per each function in terms of visual, manual and cognitive distraction. They showed that distraction interferes with the driver’s attention and the intensity and stability of his control actions [24]. 

In the current study, males 55–74-years-old were more likely to perform cell phone calls while driving and in addition to professional drivers, car drivers and urban residents were more likely to use texting and GPS setting than phone functions while driving. While professional drivers are obviously dependent on the GPS navigation aid, younger ages are widely using texting as their means of communication [25], a behavior also adopted while driving.

Simulations have shown that cell phone use for calls or texting, leads to reduced driving speed, delayed reaction time to an occurring event and a nine-fold increase in the risk for accident engagement compared to driving without cell phone. Combination of cell phone use, and other distractive factors, such as bad weather or listening to music may further increase the risk for accident [23].

Previous studies have shown that in-vehicle distractions (such as conversation with passengers, smoking, eating/drinking, presence of a pet or insect), sleepiness/daydreaming, sneezing, or engaging with vehicle equipment, greatly affects the driver’s behavior and consequently, their safety [23]. The increasing media and small screen use and browsing generate a powerful distraction from the main activity. The heavier the media multitasking, the greater the difficulty to resist distraction and focus on the primary task, and the greater the cognitive challenge posed. As for nowadays, small screens accompany almost any human activity [26,27], and their use while driving increases task difficulty and the risk of accident. 

### 4.3. 8 h Sleep Prior to Long Travel Driving 

According to the present study, more males, younger individuals and individuals with higher education reported a full night’s sleep prior to a long travel. This finding indicates the need to raise awareness and educate these diverse groups through a tailored approach on the effects of sleepiness and consequent tiredness on their driving performance. As per the ESRA2 survey [12], 25.5% of Greek participants admitted having been driving while feeling sleepy [12]. A previous study [5] found that excessive daytime sleepiness as in obstructive sleep apnea (OSA) and sleep deprivation increased the risk of truck drivers to engage in road accidents. While OSA prevalence among adults is estimated 1–6% [28], prevalence of possible OSA among professional drivers has been found to range between 15–45% [29,30]. The need for a targeted screening not only among professional but also among private drivers becomes obvious since no test, law, or traffic code of conduct is in place to identify sleep related factors in road accidents.

### 4.4. Smoking

Our findings for smoking while driving mirror the smoking prevalence by age, sex, social class and urbanization in Greece. In consonance with the 2019 smoking prevalence in Greece (36% male/22% female smokers) [14], more males, of younger age (<34-years-old), lower social class and urban residency smoke while driving, in addition to more professional than private drivers.

A Canadian study found a higher prevalence of accidents among smokers than nonsmokers, prior to the implementation of the in-car smoking ban legislation [31]. While smoking is a recognized distractive behavior and risk factor for traffic accidents, it is not widely perceived as such by drivers who may not be aware or rely on erroneous beliefs and misconceptions [32]. Almost half of the professional drivers in the present study were found to smoke and use cell phone while driving, a finding pointing to the need for a multidisciplinary address to the problem.

### 4.5. Medication Altering Concentration 

Only 1% of the current study’s participants admitted to having driven under the influence of medication causing drowsiness, while previous findings in the frame of ESRA2 survey [12], showed that 7.2% of the Greek participants admitted to having drug driven [33] a difference likely explained by the different methodology (online survey vs. CATI) and question phrasing (non-medicinal drugs in the ESRA2 vs. drowsiness causing medication in the current study). Furthermore, the same study found that alcohol, medications and drugs, are the least reported behaviors, indicating either lack of awareness, or intentional under-report. While alcohol is objectively tested using breath or blood tests by traffic controllers, irritation, tiredness and fatigue, obvious risk factors for traffic accidents, are difficult to tackle, test and scale. The same applies for therapeutic medication, recreational and addictive drugs. Furthermore, to date, there is no tool available to predict the combined effect on physical and cognitive performance exerted by the complex therapeutic regimes administered to certain population groups including but not limited to seniors. In the limitations of our study should be included, the fact that we did not specifically ask for medicinal opioids and psychoactive medication such as antidepressants and antipsychotics that were found by a previous study [6] to interfere with driving performance. 

### 4.6. Motorcycle Drivers

Motorcycle drivers, overall, reported performing less distractive behavior compared to car drivers, likely due to the intricate more challenging driving requirements and to the widely accepted vulnerability of the two-wheel drivers [13]. Two wheels are increasingly used, especially in Mediterranean countries, possibly due to the favorable climate and also due to their easier move through the congested city roads. Montella et al. [34] collected crash data from Spain and through the application of two complementary models determined the factors (arising from road, environment and drivers’ characteristics) associated with the severity and type of impact in power-two-wheel accidents. They concluded that old age (> 65 years-old) and male gender is associated with increased severity, while drivers of a very young age are more likely to engage in accidents with pedestrians. 

Previous studies have been focusing on sophisticated modelling, mathematical calculations and simulation experiments to predict the risk for accident and risk for severe injury [34,35] or to suggest the need for more restrictive enforcement, infrastructure improvement along with rider education and campaigns [13,34]. Based on video experiments and driving simulators, the theories of risk homeostasis, and risk allostasis, were formulated respectively, where participants reported their feelings of risk, task difficulty and possibility of accident. In contrast to the theory of risk homeostasis [36], theory of risk allostasis found that the abovementioned feelings were stable up to a certain threshold point and only increased once the threshold had been exceeded [35]. 

### 4.7. Future Considerations

Considering that distractive behavior adds difficulty on any given task and further complicates the challenge (by definition) of the driving act [26], the present study took a different approach; successful primary prevention is the result of actions taken prior to any risk partake or accident. Therefore, interventions should be implemented early on, to prevent the development of unsafe behaviors. In line with previous studies, current findings have shown youth to be a strong risk factor for traffic accidents [21]. As human behavior is shaped early, in the course of childhood and early adolescence, building on the current knowledge, a tailored school based educational approach will help raise a generation of conscious road users, and the ultimate means for primary prevention and public health improvement.

## 5. Conclusions

The current study is the first to focus on the factors associated with drivers’ distractive behavior in Greece. Drivers’ behavior was found to be different depending on their age, sex, social class and area of residence. Male drivers were more likely to perform drink driving, while professional drivers were more likely to use cell phone for calls and texting, set the GPS and smoke while driving. Awareness on the relation between smoking, alcohol and road accidents could be also raised through the smoking and alcohol prevention programs that in addition to those tackling driving behaviors, are expected to indirectly help reduce the risk for accidents deriving from the distraction brought up by certain lifestyle choices such as smoking while driving and drink driving. 

## Figures and Tables

**Figure 1 ijerph-18-04104-f001:**
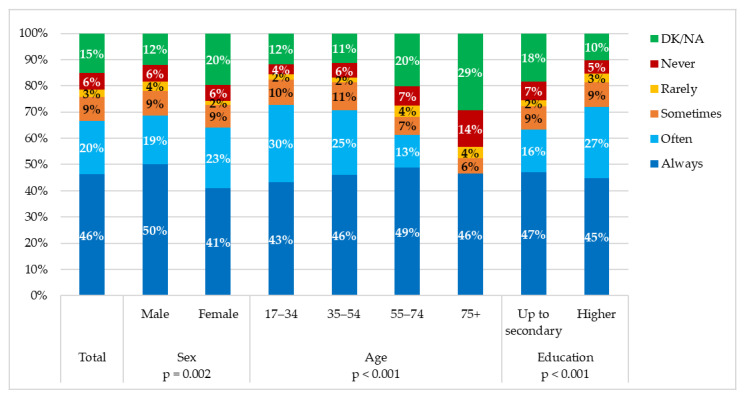
Obtaining 8-h sleep before a long journey in the total sample and by demographic characteristics of participants (question was structured as “How often do you sleep 8 h before a long journey?”).

**Table 1 ijerph-18-04104-t001:** Description of the study’s questionnaire.

Section	Theme	Participants
1	General information regarding driving	All participants with a focus on drivers
2	Violations, reactions to other driver’s irritating behavior, behavior that limits concentration while driving, vehicle maintenance, preparation before a long journey, use of child seats	Car, motorbike and professional drivers
3	Irresponsible driving behavior	Motorbike and bicycle drivers
4	Passengers’ responsibilities	All participants
5	Demographic characteristics	All participants

**Table 2 ijerph-18-04104-t002:** Demographic characteristics of study participants.

Variable	*N* = 1601
**Sex**	
Male	48%
Female	52%
**Age**	
17–34	27%
35–54	35%
55–74	29%
75+	9%
**Residence**	
Urban (>10,000 residents)	69%
Town (2000–10,000 residents)	15%
Rural (≤2000 residents)	15%
**Education**	
Up to secondary	68%
Higher	32%
**Occupational status**	
Working	42%
Unemployed	12%
Housewife	11%
Retired	26%
Student	6%
Other	2%
DK/NA *	0%
**Social class**	
Upper	15%
Middle to upper	24%
Middle to lower	38%
Lower	22%
**Driving license**	
Yes	74%
No	26%

* DK/NA: Do not know/no answer.

**Table 3 ijerph-18-04104-t003:** Characteristics of participants with at least one valid driving license.

Variable	*N* = 1178
**Number of valid driving licenses per participant**	
1	76%
2	20%
3	4%
**Valid driving license (multiple per person)**	
Car	98%
Motorbike	22%
Professional vehicle (e.g., taxi, bus, lorry etc.)	7%
**Driving years (median, IQR)**	
Car	27 (17–39)
Motorbike	23 (14–35)
Professional vehicle (e.g., taxi, bus, lorry etc.)	25 (10–33)
**Average daily driving time (minutes) (median, IQR)**	
Car	9 (7–17)
Motorbike	9 (5–17)
Professional vehicle	34 (17–69)
**Average daily driving distance (km) (median, IQR)**	
Car	7.1 (3.6–11.4)
Motorbike	5.7 (3.6–10)
Professional vehicle	28.6 (8.6–35.7)
**Most driven vehicle in the past 3 months**	
Private Car	78%
Motorbike	8%
Professional vehicle (e.g., taxi, bus, lorry etc.)	3%
Bicycle	3%
None	7%
**Ministry of Transport (MOT) test**	
Always on time	91%
Sometimes delayed	7%
DK/NA *	1%
**Gas card**	
Always valid	78%
Not always renewed on time	19%
DK/NA*	3%
**Car tyres renewal (years) (median, IQR)**	3 (3–4)
**Motorbike tyres renewal (years) (median, IQR)**	3 (2–4)

* DK/NA: Do not know/no answer.

**Table 4 ijerph-18-04104-t004:** Behavior that limits concentration in the total sample and by type of vehicle (questions were structured as “How many times in the past 3 months have you …”).

Behavior that Limits Concentration on Driving	Total	Type of Vehicle			*p*-Value
		Private Car	Motorbike	Professional Vehicle	
**Reacting to other drivers’ irritating behavior**					0.76
At least once	95%	95%	94%	97%	
Never	5%	5%	6%	3%	
**Cell phone calls**					<0.001
At least once	49%	51%	21%	63%	
Never	51%	49%	79%	35%	
**Smoking**					<0.001
At least once	20%	20%	8%	45%	
Never	79%	80%	89%	52%	
DK/NA *	1%		3%	3%	
**Setting GPS**					0.002
At least once	18%	19%	8%	19%	
Never	81%	81%	89%	76%	
DK/NA *	1%	1%	4%	4%	
**Drink driving**					0.28
At least once	13%	13%	21%	12%	
Never	87%	87%	79%	88%	
**Texting**					<0.001
At least once	10%	10%		25%	
Never	90%	89%	97%	73%	
DK/NA *	1%		3%	2%	

* DK/NA: Do not know/no answer; p-values presented derive from Chi-squared tests.

**Table 5 ijerph-18-04104-t005:** Number of valid driving licenses and mostly driven vehicle by demographic characteristics of participants.

Variable	Sex			Age					Educational Level			Social Class					Area of Residence			
	Male	Female	*p*-Value	17–34	35–54	55–74	75+	*p*-Value	Up to secondary	Higher	*p*-Value	Upper	Middle to upper	Middle to lower	Lower	*p*-Value	Urban	Town	Rural	*p*-Value
**Number of valid driving licenses per participant**			<0.001					<0.001			<0.001					<0.001				0.09
1	54%	58%		50%	65%	54%	48%		51%	68%		68%	63%	54%	43%		58%	55%	49%	
2	26%	4%		10%	19%	16%	7%		12%	19%		13%	16%	18%	9%		15%	12%	16%	
3	1%	0%		2%	4%	2%	2%		4%	19%		2%	2%	3%	4%		2%	3%	6%	
None	14%	37%		38%	12%	28%	44%		33%	12%		17%	19%	25%	44%		25%	29%	30%	
**Most driven vehicle in the past 3 months ***			<0.001					0.004			0.15					0.01				0.91
Private Car	78%	93%		78%	86%	85%	96%		83%	87%		93%	85%	82%	83%		85%	84%	84%	
Motorbike	13%	3%		11%	7%	10%	3%		9%	7%		4%	10%	9%	10%		9%	8%	8%	
Professional vehicle (e.g., taxi, bus, lorry etc.)	6%	0%		3%	4%	4%	2%		4%	2%		0%	2%	5%	5%		3%	5%	4%	
Bicycle	3%	4%		8%	3%	1%	0%		3%	4%		2%	4%	4%	2%		4%	3%	3%	

*p*-values presented derive from Chi-squared tests, * of participants owning a valid driving license.

**Table 6 ijerph-18-04104-t006:** Behavior that limits concentration by demographic characteristics of participants (questions were structured as “how many times in the past three months have you …”).

Behavior that Limits Concentration on Driving	Sex			Age					Educational Level			Social Class					Area of Residence			
	Male	Female	*p*-Value	17–34	35–54	55–74	75+	*p*-Value	Up to secondary	Higher	*p*-Value	Upper	Middle to upper	Middle to lower	Lower	*p*-Value	Urban	Town	Rural	*p*-Value
**Reacting to other drivers’ irritating behavior**			0.34					<0.001			0.07					0.03				0.64
At least once	95%	96%		99%	97%	94%	80%		95%	97%		97%	97%	95%	91%		95%	96%	94%	
Never	5%	4%		1%	3%	6%	20%		5%	3%		3%	3%	5%	9%		5%	4%	6%	
**Cell phone calls**			0.05					<0.001			0.004					0.001				0.47
At least once	51%	46%		49%	59%	43%	11%		45%	55%		50%	49%	54%	35%		50%	51%	43%	
Never	48%	54%		51%	41%	56%	89%		55%	45%		50%	51%	45%	65%		50%	49%	57%	
**Smoking**			0.13					0.0005			0.15					0.02				0.06
At least once	20%	19%		16%	26%	17%	4%		21%	18%		13%	19%	20%	29%		22%	21%	11%	
Never	78%	81%		83%	73%	83%	95%		78%	82%		87%	81%	79%	71%		78%	78%	88%	
DK/NA*	1%			2%	1%		1%		1%				1%	1%					2%	
**Setting GPS**			0.35					<0.001			0.0009					0.18				0.03
At least once	18%	18%		33%	20%	7%	1%		15%	22%		17%	21%	19%	13%		20%	18%	9%	
Never	81%	82%		66%	80%	91%	95%		83%	78%		83%	79%	80%	85%		79%	82%	88%	
DK/NA *	1%			1%		2%	4%		1%					1%	2%		1%	1%	3%	
**Drink driving**			<0.001					0.08			0.33					0.79				0.39
At least once	20%	5%		17%	12%	15%	3%		13%	13%		12%	13%	15%	11%		14%	11%	11%	
Never	80%	95%		83%	88%	85%	97%		87%	87%		87%	86%	85%	89%		86%	89%	88%	
**Texting**			0.25					<0.001			0.0002					0.02				0.01
At least once	9%	11%		23%	10%	2%			7%	14%		7%	7%	14%	7%		11%	8%	5%	
Never	90%	89%		76%	90%	97%	98%		92%	85%		93%	93%	85%	92%		88%	91%	93%	
DK/NA *	1%			1%		1%	2%		1%					1%	1%			1%	2%	

* DK/NA: Do not know / no answer, *p*-values presented derive from Chi-squared tests.

**Table 7 ijerph-18-04104-t007:** Logistic regression models for drivers’ characteristics that affect behavior that limits concentration while driving.

Model	Explanatory Variables		Odds Ratio	*p*-Value	95% Confidence Interval	
					Lower	Upper
Dependent variable: **Drink driving**	**Sex** (reference female)	Male	5.04	<0.001	3.12	8.14
	**Age** (reference 75+)	17–34	7.97	0.002	2.13	29.84
		35–54	6.21	0.005	1.74	22.10
		55–74	6.82	0.003	1.92	24.13
		Constant	0.008	<0.001	0.002	0.03
Dependent variable: **Cell phone calls**	**Sex** (reference female)	Male	1.18	0.656	0.58	2.41
	**Age** (reference 17–34)	35–54	1.18	0.580	0.66	2.12
		55–74	0.40	0.005	0.21	0.76
		75+	0.04	0.003	0.005	0.32
	**Vehicle** (reference motorbike)	Car	5.64	<0.001	3.07	10.35
		Professional vehicle	6.78	<0.001	2.59	17.71
	**Age ## Sex**	Male 35–54	1.53	0.309	0.67	3.50
		Male 55–74	2.75	0.019	1.18	6.41
		Male 75+	3.08	0.330	0.32	29.73
		Constant	0.18	<0.001	0.09	0.40
Dependent variable: **Smoking**	**Sex** (reference female)	Male	1.43	0.043	1.01	2.04
	**Age** (reference 17–34)	35–54	1.99	0.009	1.186	3.33
		55–74	1.04	0.876	0.60	1.81
		75+	0.17	0.006	0.05	0.61
	**Vehicle** (reference motorbike)	Car	3.56	0.002	1.61	7.91
		Professional vehicle	9.53	<0.001	3.25	28.00
	**Social class** (reference upper)	Middle to upper	1.64	0.074	0.95	2.82
		Middle to lower	1.62	0.068	0.97	2.72
		Lower	3.73	<0.001	2.06	6.78
	**Area of residence** (reference rural)	Urban	2.95	0.002	1.47	5.92
		Town	2.69	0.013	1.23	5.88
		Constant	0.01	<0.001	0.003	0.03
Dependent variable: **Texting and/or setting the GPS**	**Sex** (reference female)	Male	1.30	0.153	0.91	1.86
	**Age *** (reference 55–74)	17–34	9.10	<0.001	5.51	15.02
		35–54	3.99	<0.001	2.64	6.05
	**Vehicle** (reference motorbike)	Car	5.01	0.003	1.75	14.32
		Professional vehicle	9.24	0.001	2.46	34.72
	**Area of residence** (reference rural)	Urban	2.66	0.008	1.29	5.48
		Town	2.02	0.096	0.88	4.64
		Constant	0.01	<0.001	0.002	0.03

All models were concluded using backward elimination procedure. Variables considered were sex, age and all variables statistically significant in the univariate analyses, * The age category 75+ was excluded from this analysis due to a very small sample count.

## Data Availability

The data presented in this study are available in Zenodo (10.5281/zenodo.4683094).

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
