# Peer review of "Driving Behavior That Limits Concentration: A Nationwide Survey in Greece"

_ijerph, 2021, doi:10.3390/ijerph18084104_

Round 1
Reviewer 1 Report
This is an interesting paper that examines the driver's behaviour and factors affecting driver's concentration in Greece. The paper is well structured and written, the presented methodology is scientifically sound and successfully conveyed to the reader. Finally, the conclusions are valid and well supported by the paper. A few comments are suggested to be addressed before publication:
- A pan European survey (https://www.esranet.eu/en/) has been conducted and results on self-reported behaviour concerning key risk factors have been published for Greece as well. Related publications follow, that are recommended to be included in the paper. It would also be interesting to compare the results of the two surveys concerning the common risk factors examined (e.g. drink driving, use of mobile phone):
- Yannis G., Nikolaou D., Laiou A., Achermann Stürmer Y., Buttler I., Jankowska-Karpa D., Vulnerable road users: Cross-cultural perspectives on performance and attitudes, IATSS Research, Volume 44, Issue 3, 2020, Pages 220-229, ISSN 0386-1112, https://doi.org/10.1016/j.iatssr.2020.08.006.
- Pires C., Torfs K., Areal A., Goldenbeld C., Vanlaar W., Granié M.A, Achermann Stürmer Y., Shingo Usami D., Kaiser S., Jankowska-Karpa D., Nikolaou D., Holte H., Kakinuma T., Trigoso J., Van den Berghe W., Meesmann U., Car drivers' road safety performance: A benchmark across 32 countries, IATSS Research, Volume 44, Issue 3, 2020, Pages 166-179, ISSN 0386-1112, https://doi.org/10.1016/j.iatssr.2020.08.002.
- A better formatting and overall presentation of the tables is recommended.
Reviewer 2 Report
- The design of Table 3 is confusing
- It is suggested to classify all these factors. For example, which variables are used to characterize driving behavior?
- I think the part of the questionnaire about driving habits (section 2) is more interesting, but this part is missing in the result analysis.
- Is there any traffic crash data in the study? What are the dependent variables of logistic regression?
- The conclusion, such as male drivers are more likely to drink and skilled drivers are more likely to use mobile phones, is not innovative.
- The BMA approach can overcome the model uncertainty for analyzing the impact of variables on driving behaviors, and should be reviewed and discussed in the manuscript. For example, see Chen, Y., Persaud, B., & Sacchi, E. (2012). Improving transferability of safety performance functions by Bayesian model averaging. Transportation research record, 2280(1), 162-172. Zou, Y., Lin, B., Yang, X., Wu, L., Abid, M., and Tang, J., Application of the Bayesian model averaging in analyzing freeway traffic incident clearance time for emergency management, Journal of Advanced Transportation, 2021, in press.
Reviewer 3 Report
The article is very interesting as it studies indirect road safety measures such as driving behavior, using a database of 1601 participants.
Its strength is the use of innovative data. However, there are two critical points:
- A surrogate measure of safety to be such, respect the following requirement: a variation of this measure must correspond to a variation in the number of accidents and in the severity of these. I understand that it is difficult to demonstrate this association, so the authors must expand the state of the art by introducing articles that can support such correspondences. Among the articles that can help authors who can help are:
- Tarko, A. (2019). Measuring road safety with surrogate events. Elsevier.
- Paleti, R., Eluru, N., & Bhat, C. R. (2010). Examining the influence of aggressive driving behavior on driver injury severity in traffic crashes. Accident Analysis & Prevention, 42(6), 1839-1854.
- Amoh-Gyimah et al. 2016 (Macroscopic modeling of pedestrian and bicycle crashes: A cross-comparison of estimation methods) studied how the socioeconomic deprivation level influenced the number of road accidents.
- Montella et al. 2020 (A data mining approach to investigate patterns of powered two-wheeler crashes in Spain) have identified the correlation with gender and age.
- Fuller proposed the “task-capability” (TCI) model, which describes the interaction of the main factors influencing driver behavior and provides a dynamic control of the motivational framework of drivers' actions (Fuller, R., 2005. Towards a general theory of driver behavior Accident Analysis and Prevention Vol. 37, pp. 461–472; Fuller, R., McHugh, C., Pender, S., 2008. Task difficulty and risk in the determination of driver behavior. Revue européenne de psychologie appliquée Vol. 58, pp. 13–21.).
- The second critical point concerns questionary validation. In this case, the data are the result of a collection of information through the questionnaires. Since very often the subjects respond without paying great attention, especially with a questionnaire as large as this one (consisting of 43 questions), it is necessary to validate this questionnaire. The procedure is very long, in particular it is advisable to estimate the Cronbach coefficient. For each question administered, the correlation and the respective p-value are calculated. The questions that need to be analyzed are only those that are statistically significant.
- The analysis methodology is appropriate, however mixed logit could be used instead of logit, as it allows to analyze also phenomena of heterogeneity of the data, having a very extensive sample of analysis.
In conclusion, the article has a good foundation, I personally think that the state of the art should be strengthened. The point that absolutely needs to be improved is the correctness of the data. In all guides for reviewers, it is essential to evaluate the appropriateness of the data.
Round 2
Reviewer 2 Report
None.
Reviewer 3 Report
Thanks to the authors who tried to answer all the questions asked. I think the article can be accepted